# Neuroprotective Effects of Transferrin in Experimental Glaucoma Models

**DOI:** 10.3390/ijms232112753

**Published:** 2022-10-22

**Authors:** Jenny Youale, Karine Bigot, Bindu Kodati, Thara Jaworski, Yan Fan, Nana Yaa Nsiah, Nathaniel Pappenhagen, Denise M. Inman, Francine Behar-Cohen, Thierry Bordet, Emilie Picard

**Affiliations:** 1Eyevensys, 11 Rue Watt, 75013 Paris, France; 2Centre de Recherche des Cordeliers, INSERM, Université de Paris Cité, Sorbonne Université, From Physiopathology of Ocular Diseases to Clinical Development, 75006 Paris, France; 3Department of Pharmaceutical Sciences, North Texas Eye Research Institute, University of North Texas Health Science Center, Fort Worth, TX 76107, USA; 4Cochin Hospital, AP-HP, Assistance Publique Hôpitaux de Paris, 24 rue du Faubourg Saint Jacques, 75014 Paris, France

**Keywords:** ferroptosis, glaucoma, iron, neuroprotection, ocular hypertension, RGC, retinal degeneration, transferrin

## Abstract

Iron is essential for retinal metabolism, but an excess of ferrous iron causes oxidative stress. In glaucomatous eyes, retinal ganglion cell (RGC) death has been associated with dysregulation of iron homeostasis. Transferrin (TF) is an endogenous iron transporter that controls ocular iron levels. Intraocular administration of TF is neuroprotective in various models of retinal degeneration, preventing iron overload and reducing iron-induced oxidative stress. Herein, we assessed the protective effects of TF on RGC survival, using ex vivo rat retinal explants exposed to iron, NMDA-induced excitotoxicity, or CoCl_2_-induced hypoxia, and an in vivo rat model of ocular hypertension (OHT). TF significantly preserved RGCs against FeSO_4_-induced toxicity, NMDA-induced excitotoxicity, and CoCl_2_-induced hypoxia. TF protected RGCs from apoptosis, ferroptosis, and necrosis. In OHT rats, TF reduced RGC loss by about 70% compared to vehicle-treated animals and preserved about 47% of the axons. Finally, increased iron staining was shown in the retina of a glaucoma patient’s eye as compared to non-glaucomatous eyes. These results indicate that TF can interfere with different cell-death mechanisms involved in glaucoma pathogenesis and demonstrate the ability of TF to protect RGCs exposed to elevated IOP. Altogether, these results suggest that TF is a promising treatment against glaucoma neuropathy.

## 1. Introduction

Glaucoma is the second leading cause of global blindness with up to 11 million affected people worldwide in 2020 [1]. While current treatments reduce intraocular pressure (IOP), none of them are currently approved for delaying long-term optic nerve degeneration (i.e., glaucomatous neuropathy). Glaucomatous neuropathy is characterized by the progressive loss of retinal ganglion cells (RGCs) and optic nerve axons, resulting from intricate mechanisms including inflammation [2], elevated intraocular pressure, oxidative stress [3], retinal ischemia [4], glutamate excitotoxicity [5], apoptosis [6], necrosis [7], and disruption of iron homeostasis [8]. 

Iron is one of the most abundant metals in the retina and is essential for retinal function, acting as a co-factor of key retinal enzymes involved in the phototransduction cascade, photoreceptor outer segment disc biogenesis, energy production, etc. (recently reviewed in [8]). In the retina, labile iron content is tightly regulated. While the blood–retinal barrier protects the retina from systemic iron loading, iron intake from choroidal circulation across the retinal pigment epithelium (RPE) is allowed by apo-transferrin (TF), an endogenous iron transporter. In the retina, iron overload promotes the formation of reactive oxygen species through the Fenton and Haber–Weiss reactions, and activates many genes involved in oxidative metabolism and stress through its binding to the IRE (iron-responsive element) regulatory sequence. Iron-induced damage, including oxidative stress and cell death, occur through several mechanisms including ferroptosis that results from iron-mediated lipid peroxidation [9]. 

Altered iron homeostasis was evidenced in animal models and in glaucomatous patients. Iron-regulating proteins such as TF, ceruloplasmin, and ferritin are upregulated after optic nerve crush in rats [10], and are higher in the aqueous humor [11] and retina [12,13] of patients and of cynomolgus monkeys suffering from glaucoma, suggesting a role for iron-induced oxidative stress in glaucoma pathogenesis. Whether iron accumulation is a cause or consequence of glaucoma remains elusive. However, several lines of evidence indicate that iron might directly contribute to glaucoma pathogenesis. Iron overload exaggerated the loss of RGCs after partial optic nerve crush in rats [14] and chemical metal chelator ameliorated neurodegeneration in rat eyes with elevated IOP [15]. Iron-chelating agents protected RGCs against N-methyl-D-aspartate (NMDA)-induced excitotoxicity [16] and against microbead-induced hypertension [17]. The risk of glaucoma was higher in individuals with high serum ferritin levels [18], and in those who ingested high levels of iron and calcium supplements [19]. Iron chelation was thus identified as a promising therapeutic strategy for retinal degeneration, including glaucoma [20].

As compared to chemical iron chelators, TF presents several advantages. It is an endogenous iron-regulatory protein, without risk of excessive iron chelation or direct toxic effects as observed with chemical chelators [8]. We previously demonstrated that overexpression of human TF in transgenic mice, or intraocular delivery of TF, is safe and neuroprotective in various models of retinal degeneration, reducing retinal iron content, controlling iron-induced oxidative stress, slowing down photoreceptor death, and preserving function [21,22,23,24,25]. Results also suggest that TF possibly regulates other iron unrelated pathways [24]. 

There is no animal model that fully recapitulates human glaucomatous neuropathy, which is still imperfectly understood, but several models are used to mimic ganglion cell death resulting from mechanisms identified in the pathogenesis of the disease [2,3,4,5,6,7,8,26]. Herein, we have investigated the neuroprotective potential of TF in various experimental conditions mimicking pathways involved in glaucoma neuropathy, including excitotoxicity [5], hypoxia [26], and iron intoxication [8], using ex vivo organotypic rat retinal explants. TF effects on RGC survival were also assessed in the pathogenic context of elevated intraocular pressure (IOP) using the in vivo microbead rat model of ocular hypertension. Our results suggest that TF could be a promising candidate for the treatment of glaucomatous neuropathy through multiple neuroprotective mechanisms. 

## 2. Results

### 2.1. TF Protects RGCs from Iron Intoxication

Because glaucomatous neuropathy is thought to result from intricate mechanisms including hypoxia, excitotoxicity, and iron dysmetabolism, we submitted rat retinal explants to several stresses and explored if TF could protect RGCs. Explants were subjected to 1 mM FeSO_4_ for 24 h and maintained in culture until 96 h in the presence of TF at 50 mg/mL, a concentration previously shown to protect retinas ex vivo [24]. At the end of the incubation period, RGCs were immunolabeled with anti-Brn3a antibody, a specific marker for RGCs, and counted. Iron induced a significant decrease in RGC number (46.7 % reduction, *p* < 0.0001; Figure 1A,B) and a drastic increase in lactate dehydrogenase (LDH) release (*p* < 0.001, Figure 1C), attesting to extensive cell membrane damage. Treatment with TF significantly reversed iron-induced toxicity, preserving 40.6 % of the RGCs when compared to untreated FeSO_4_-exposed explants (*p* = 0.0121; Figure 1B), whereas TF applied alone to the explant did not modify RGC survival [24]. TF also precluded FeSO_4_-induced LDH release (Figure 1C), demonstrating preservation of cell integrity.

### 2.2. TF Protects RGCs from NMDA-Induced Excitotoxicity and From CoCl_2_-Induced Hypoxia

We first confirmed that RGCs were sensitive to excitotoxicity induced by the glutamate receptor agonist NMDA (Appendix A) and to hypoxia induced by cobalt chloride (CoCl_2_) (Appendix A), with a dose-dependent loss of RGCs when stresses were applied for 24 h and explant cultures extended up to 96 h total. Under these conditions, photoreceptors were spared by the two death-inducers (Appendix A). When applied at 100 µM, NMDA led to a significant decrease in the number of RGCs (53.0%, *p* < 0.0001 compared to control; Figure 2A,B) correlating with an increase in LDH release (Figure 2C). TF significantly prevented part of the NMDA-induced RGC loss (53.5% compared to NMDA-exposed explants, *p* = 0.0004; Figure 2B) and reduced LDH release (Figure 2C). Similarly, 100 µM CoCl_2_ led to a significant loss of RGCs (45.7%, *p* < 0.0001 compared to control; Figure 2D–E) which was prevented by TF (37.1% compared to CoCl_2_-exposed explants, *p* = 0.0072). It is of note that no significant LDH release was observed after CoCl_2_ exposure, suggesting a non-necrotic cell-death pathway could be involved in RGC loss. These data indicate that TF could prevent RGC losses from glutamate excitotoxicity and from hypoxic conditions.

### 2.3. TF Protects RGCs from Different Cell-Death Mechanisms

To gain an understanding of the neuroprotective mechanism of action with TF, the death mechanisms induced by each of the three stresses were defined after 24 h incubation and at the end of the culture period (24 h stress + 72 h extended culture). Terminal deoxynucleotidyl transferase (TdT)-mediated dUTP nick-end labeling (TUNEL)-positive cells were quantified in the ganglion cell layer of explants as a marker of apoptosis. Receptor-interacting protein kinase 3 (RIP3) and Glutathione peroxidase 4 (GPX4) immunostaining intensities in the ganglion cell layer were quantified on explant sections as markers of necrosis and ferroptosis, respectively (Appendix A). 

Exposure to FeSO_4_ induced a significant increase in GPX4 expression at 24 h (*p* = 0.0004 compared to control) and a progressive increase in RIP3 immunostaining at 24 h and 96 h, while the number of TUNEL-positive cells was not markedly changed (Figure 3A), showing both ferroptotic and necrotic cell death of RGCs from FeSO_4_ exposure. NMDA first elicited an increase in GPX4 expression (*p* = 0.0032 at 24 h) accompanied by an increase in the number of TUNEL-positive cells. There was also a progressive increase in RIP3 expression, reaching about a six-fold level at 96 h compared with control levels (Figure 3B), indicating that ferroptosis, apoptosis, and necrosis could be involved in RGC loss after NMDA exposure. Finally, CoCl_2_ had no marked effect on RIP3 or GPX4 expression while progressively increasing the number of TUNEL-positive cells (*p* = 0.0135 and *p* < 0.0001 at 24 and 96 h, respectively; Figure 3C), suggesting an apoptotic cell-death mechanism. These data indicate that at the ganglion cell layer, iron exposure mainly induces ferroptosis and necrosis, CoCl_2_-induced hypoxia mainly leads to apoptosis, and NMDA-mediated excitotoxicity results in activation of all three death pathways. 

Treatment with TF not only reduced the increase in the number of TUNEL-positive cells induced by NMDA and CoCl_2_ (Figure 3B,C), but also reduced and even prevented the increase in RIP3 immunostaining induced by FeSO_4_ and NMDA, respectively (Figure 3A,B). Finally, TF prevented the increase in GPX4 expression induced by FeSO_4_ or NMDA, maintaining control levels of GPX4 in both conditions (Figure 3A,B). Altogether, these findings indicate that the protective effect of TF is not limited to ferroptosis but extends to other death mechanisms such as apoptosis or necrosis.

### 2.4. TF Prevents RGC Death and Axonal Loss in Rats Subjected to Increased IOP

To determine the potential neuroprotective role of TF against RGC degeneration in vivo, we investigated its effect in a validated animal model for glaucoma: the microbead occlusion model in rats. Indeed, drainage angle occlusion with magnetic microbeads has been used to generate rapid and stable increases in IOP in rodents leading to progressive RGC degeneration and axonal loss at the optic-nerve level [27]. Herein, rats were randomized into three groups based on baseline IOP (13.8 ± 0.18 mmHg average IOP for all rats; Appendix A). 

Following microbead injection, IOP quickly increased and remained elevated in all microbead-injected eyes during the 4 weeks of the experimental period compared to normotensive eyes (Figure 4A). As expected, IOP remained unchanged in the naïve rats that were not subjected to OHT injury. Importantly, IOP in the microbead-injected eyes of the vehicle group and of the TF group were not significantly different at any point during the study (Figure 4A and Appendix A). After last IOP measurements on day 28, rats were euthanized, and retinas and optic nerves were collected for RGC counting on flat-mount RPBMS immunostaining (Figure 4B) and axon quantification on optic nerve cross-sections. As expected, vehicle OHT controls had 50.3% fewer RGCs than naïve controls (958 ± 146 and 1927 ± 61 cells/mm^2^, respectively; *p* < 0.0001), confirming successful RGC loss by induction of OHT (Figure 4C). In comparison, RGC density in the TF group was not different from the naïve control group (*p* = 0.2475), and thus, TF administration preserved RGC by 70.5% compared to vehicle-treated animals with OHT (1634 ± 83 vs. 958 ± 146 cells/mm^2^, *p* = 0.0008). The axon density in optic nerve cross-sections was also significantly reduced in vehicle OHT controls compared to naïve controls (*p* < 0.0001; Figure 4D). Axon density in the TF group was not different from the naïve control group (310,134 ± 24,250 vs. 353,962 ± 11,317 axon/mm^2^, respectively; *p* = 0.2480). TF administration preserved axon density by 46.9% compared to vehicle-treated animals with OHT (310,134 ± 24,250 vs. 211,142 ± 18,960 axon/mm^2^, respectively; *p* = 0.0027). Overall, these results indicate that TF can prevent RGC degeneration due to an elevated IOP in rats.

### 2.5. Iron Overload in the Retina of Human Glaucomatous Eyes

To confirm iron accumulation in human glaucomatous retinas, glaucoma and healthy retina sections were stained with enhanced Perls’ reaction (Figure 5). In the mid- and peripheral retinas of healthy patients, iron deposits were only observed in cone outer segments (arrow) and in the Müller glial cell end-feet (arrowhead) (Figure 5A3). In retinas from a glaucomatous patient, brown iron precipitates were observed in the optic nerve (Figure 5B1, asterisk) in the nerve fiber layer and in the ganglion cells around the optic nerve head (Figure 5B2, thin arrows), whilst iron labeling was reduced in cone outer segments and higher in Müller glial cell end-feet (Figure 5B3) compared to healthy retinas. In the peripheral retina from the glaucomatous eye (Figure 5B4), iron accumulated in Müller glial cells (compared to healthy eye Figure 5A4). These results show that iron distribution differs in glaucomatous as compared to healthy retinas, with a reduction in cones and overload in Müller glia, RGCs, and axons surrounding the optic nerve.

## 3. Discussion

The chemical properties of iron place it at the heart of many pathophysiological processes involved in neurodegenerative diseases such as oxidative stress, inflammation, hypoxia, excitotoxicity, and ferroptotic cell death [8]. Indeed, iron has been suggested to play a critical role in the progression of many retinal degenerative diseases such as age-related macular degeneration, retinal detachment, retinitis pigmentosa, diabetic retinopathy, and glaucoma [8]. 

The multifactorial dimension of glaucoma compels the investigation of neuroprotective agents available to protect against several death pathways since apoptosis, necrosis, and ferroptosis were described in RGCs from human glaucomatous retinas [28]. Chemical iron chelators efficiently reduced RGC death in an NMDA-induced excitotoxicity model in rodents [16] and in the microbead-induced hypertensive glaucoma model [17]. However, chemical chelators might not have the ability to protect against all cell-death mechanisms and were shown to expose subjects to several ocular side-effects, including vision loss and neuropathy [29,30]. Excessive retinal iron depletion by chemical chelators might also compromise the normal function of vision, as most of the enzymatic reactions involved in phototransduction require iron as a co-factor. We previously showed in multiple retinopathy models that a local intraocular TF administration preserved both retina morphology and function by controlling iron-induced oxidative stress and through iron-unrelated pathways, without any side-effects [8]. 

To provide evidence that TF has potential in the treatment of glaucoma neuropathy, we have explored its protective effects on RCGs in various ex vivo as well as in vivo relevant models. Our results showed that iron intoxication induced a significant increase in necrotic and ferroptotic death markers in the RGC layer. Ferroptosis contributes to RGC death due to the dysregulation of iron and its toxic accumulation. NMDA-mediated excitotoxicity induced necrosis and apoptosis as previously described [6], but additionally induced ferroptosis. Indeed, stimulation of the NMDA receptor induces iron uptake through Dexras1- divalent metal transporter 1 (DMT1) interaction, leading to activation of the iron import channel, DMT1 [31,32].. Cobalt (II) chloride (CoCl_2_), commonly used to mimic hypoxia-mediated oxidative stress, led to RGC death by apoptosis as previously reported in porcine retina [33]. Our results do not support the involvement of ferroptosis in CoCl_2_-induced RGC loss. Considering these results, these three models proved to be relevant to the study of the neuroprotective mechanisms of TF against the different cell-death processes initiated in RGCs. TF treatment was efficient in preserving RGCs from iron-induced necrosis and ferroptosis, NMDA-induced necroptosis, apoptosis and ferroptosis, and CoCl_2_-induced apoptosis, demonstrating the strong and multiple neuroprotective mechanisms of TF and its high potential in glaucoma neuropathy treatment since apoptosis, necrosis, and ferroptosis were described in RGCs from human glaucoma retinas [28].

As intraocular pressure is a major factor associated with glaucoma, it was important to demonstrate that TF could also protect the retina from hypertension-induced neuropathy. The in vivo microbead model in rodents is recognized as a valid model, as it induces a progressive degeneration of the RGCs and their axons in the optic nerve consecutively to the rapid and stable increase in IOP [34]. In this model, we showed that intraocular administration of TF effectively preserves RGCs and their axons from IOP-induced degeneration. RGC density and axon density in TF-treated rats exhibited similar levels to those defined in naïve rats, strongly supporting its therapeutic potential. As glaucoma is a progressive and chronic disease, long-term treatment should be anticipated. While repeated intravitreal injection is not a viable approach for patients, we have developed and demonstrated the feasibility of non-viral gene delivery for the sustained intraocular production of TF using ciliary muscle electroporation [25]. Further studies in humans are needed to evaluate this potential.

The link between iron and glaucoma is not yet clearly established. Numerous studies have reported altered iron homeostasis in glaucomatous patients (Table 1). In serum, higher TF, ferritin, and iron levels—together with lower ceruloplasmin levels—were measured in glaucomatous patients as compared to control individuals 40,42,43,45,49,18]. In both monkey and human glaucomatous retinas, mRNA and proteins of ceruloplasmin, ferritin, and transferrin were increased [12,13,35]. Besides being localized to Müller glial cells, ceruloplasmin and TF show increased immunostaining in the periphery compared to central retina [12,35]. In the aqueous humor, TF and hepcidin levels were increased in primary open-angle glaucoma (POAG) eyes as compared to control eyes undergoing cataract surgery [11,36], suggesting a deregulation of iron homeostasis in glaucomatous conditions. Variable results have been published regarding the levels of free iron in the ocular media. Iron levels were increased in the tears of glaucomatous patients [37]. However, whilst low iron levels were observed in aqueous humor from pseudo-exfoliating glaucoma (PEXG) patients [38] and high levels in POAG [39,40], in one recent study, no difference between PEXG or POAG patients compared to control was reported [41]. Saturation of transferrin and measurement of other iron-regulating proteins might be more relevant than the sole measurement of iron to better identify iron homeostasis deregulation. 

We observed accumulation of iron in the optic nerve, axons, and RGCs together with reduced iron in the inner segments of cone photoreceptors of glaucomatous eyes. Iron deposits were also observed along Müller glial cells in the peripheral retina from glaucomatous eyes, which reinforces the hypothesis that Müller glia could play a central role in iron trafficking, as previously shown by other authors who found elevated TF, ferritin, and ceruloplasmin staining in Müller cells [12,13,35]. Garcia-Castineiras et al. proposed a model of iron outflow from the eye, starting from the egress of iron from the retina through Müller glial cells toward lens epithelial cells, up to the aqueous humor where iron is thought to be recycled in circulation or lacrimal fluids [47]. Indeed, high iron and ferritin serum and lacrimal fluid levels were measured in glaucoma patients (Table 1). 

The mechanism of iron overload in glaucoma is not fully understood. Iron homeostasis dysregulation could be related at least in part to subclinical inflammation, chronic mechanical and vascular stress that alters the blood ocular barriers [48,49]. Another mechanism could involve mitochondrial dysfunction or altered transport along axons and glial Müller cells [6], as mitochondria play a major role in iron metabolism and are involved in ferroptosis [50]. This hypothesis is supported by some forms of congenital glaucoma associated to optineurin mutations that interfere with mitophagy and autophagy [51]. Recently, autophagy and mitophagy have been associated with iron metabolism deregulation [52,53]. It can, thus, not be excluded that deregulation of iron metabolism could be an early event in glaucomatous neuropathy and not only secondary to cell injury/death and/or chronic inflammation. In this sense, a direct link between ocular hypertension and iron accumulation has been recently reported in mice where rapid retinal iron accumulation was observed within 1 to 8 h after induction of pathologically high intraocular pressure. Disturbance of iron metabolism in this model was associated with nuclear receptor coactivator 4 (NCOA4)-mediated degradation of ferritin [46]. Although there is no published data reporting a direct link between normotensive glaucoma (NTG) and intraretinal iron accumulation, some evidence suggests a dysregulation of iron metabolism in normotensive glaucoma, too. Indeed, mutations in the optineurin (OPTN) gene have been identified as a causative factor for NTG [54,55], and were shown to induce a disruption in Rab8 and TF receptor 1 (TFR1) interaction leading to a defect in recycling of the TFR1/TF complex, which is essential for the control of iron levels [56,57]. In addition, normotensive glaucoma is a multifactorial disease also characterized by vascular dysregulation and increased endothelin-1 (ET-1) content [58,59]. Interestingly, increased ET-1 levels lead to iron accumulation [60]. This suggests that the neuroprotective role of TF could be extended to different types of glaucoma including normotensive and hypertensive forms.

## 4. Materials and Methods

### 4.1. Animals

Male Wistar rats, used for retinal explants, were purchased from Janvier Labs (Le Genest Saint-Isle, France). Male Brown Norway rats, used in the microbead occlusion model, were obtained from Charles River Laboratories (Charles River Laboratories, Wilmington, MA, USA). Animals were housed in a 12:12 light–dark cycle with ad libitum access to food and water. All experiments were performed in accordance with the ARVO (Association for Research in Vision and Ophthalmology) statement for the Use of Animals in Ophthalmic and Vision Research.

### 4.2. Retinal Explant Ex Vivo Models

#### 4.2.1. Retinal Explant Culture

Adult Wistar rat retinas were dissected radially into four equal-sized explants, and individually positioned onto 0.2 µm polycarbonate membranes (Merck, Saint Quentin Fallavier, France) with the RGC side facing up. The membranes were placed into the wells of a 6-well plate in Neurobasal-A (ThermoFischer Scientific, Courtaboeuf, France) supplemented with 10% decomplemented fetal bovine serum (ThermoFischer Scientific), 0.8 mM L-glutamine (ThermoFischer Scientific), 1% penicillin and streptomycin (ThermoFischer Scientific), and 0.1% fungizone (ThermoFischer Scientific). Retinal explant cultures were maintained in humidified incubators at 37 °C and 5% CO_2_. RGCs were subjected to iron intoxication, excitotoxicity, or hypoxia by addition of 1 mM FeSO_4_ (Merck), 100 µM N-Methyl-D-aspartic acid (NMDA, Merck), or 100 µM cobalt chloride (CoCl_2_, Merck), respectively, for 24 h, and then washed and further cultured in complete neurobasal medium until 96 h in total (n = 4–10 explants per condition). Human apo-transferrin (50 mg/mL; Merck) was added to the culture medium during stress-induction and left for the whole culture period (96 h). 

#### 4.2.2. RGC Quantification

Rat retinal explants were fixed with 4% paraformaldehyde (PFA; Inland Europe, Conflans sur Lanterne, France) in phosphate buffer saline solution (PBS) for 20 min, then permeabilized and blocked, respectively, in 1% Triton and 1% BSA/0.1% Triton in PBS. RGCs were immunostained with goat anti-rat Brn3a (1:300, sc31984, Santa Cruz Biotechnologies, Heidelberg, Germany) antibodies and revealed with Alexa Fluor 488–conjugated donkey anti-goat IgG (1:200, ThermoFischer Scientific) secondary antibodies. Nuclei were counterstained using 4.6-diamidino-2-phenylindole DAPI (Merck). For each explant, 3 to 5 standardized photomicrographs, similarly distributed for all conditions to cover central to peripheral retina, were taken at 40× magnification under a fluorescence microscope (BX51, Olympus, Rungis, France). Quantification of Brn3a-positive RGCs was performed on each image by an operator masked for the treatment. The number of RGCs was reported as the mean number of cells per field (photomicrograph). 

#### 4.2.3. LDH Assay

To assess cell viability, the LDH assay was performed according to manufacturer’s instructions (Merck). Briefly, 100 µL of culture medium were collected at 3 h and 24 h after intoxication and incubated with 100 µL of Cytotoxicity Detection Kit reaction mixture for 20 min at room temperature in the dark. The optical density of the solution was measured on an ELISA plate reader using 492 and 620 nm filters (Benchmark Plus microplate spectrophotometer, BioRad, Marnes-la-Coquette, France). The LDH release between the 2 time points was calculated by subtracting the absorbance measured at 24 h from the absorbance measured at 3 h and normalized to the absorbance measured in control explants without treatment.

#### 4.2.4. Immunofluorescence Staining

Rat retinal explants were fixed with 4% PFA (Merck) and snap-frozen in OCT (TissueTek, Merck,). Ten µm thick cryosections were prepared and permeabilized using 0.1% Triton X-100 (Merck). Non-specific staining was blocked in 1% Bovine Serum Albumin (Merck)/0.1% Triton in PBS for 30 min at room temperature followed by immunostaining using primary antibodies as described in Table 2, and Alexa Fluor 647–conjugated donkey anti-goat IgG or Alexa Fluor 596–conjugated donkey anti-goat IgG (1:200, ThermoFischer Scientific) secondary antibodies. Nuclei were counterstained with DAPI. Photographs were acquired with a fluorescence microscope using identical exposure parameters for all compared samples (Olympus BX51 or Confocal Zeiss LSM 710). Images were processed with ImageJ software. The RIP3 and GX4 fluorescence intensity was quantified in RGC areas with associated DAPI staining. Results were expressed as fold change versus baseline fluorescence intensity quantified on fresh retinas (n= 4–10 per group).

#### 4.2.5. TUNEL Assay

The terminal deoxynucleotidyl transferase dUTP nick-end labeling (TUNEL) assay was performed according to manufacturer’s instructions (Merck). Briefly, cryosections of retinal explants were washed with PBS and incubated for 2 min with 0.1% Triton X-100 in 0.1% sodium citrate on ice. Then, sections were incubated for 60 min at 37 °C with the reaction mixture (TUNEL enzyme plus TUNEL label [1/9]), and finally, nuclei were counterstained with DAPI. Photographs were acquired with a fluorescence microscope (Olympus BX51) using identical exposure parameters for all compared samples. Images were processed with ImageJ software. The TUNEL-positive nuclei were quantified in RGC areas with associated DAPI staining. The number of TUNEL-positive cells in a retinal explant was counted in five random fields on each retinal explant (n = 4–10 per group), and the average number of TUNEL-positive cells per random field was determined for each condition.

### 4.3. Microbead Occlusion Model

#### 4.3.1. Microbead Injection and IOP Measurements

The microbead occlusion model was used to induce ocular hypertension as previously described [61] with the following adaptations. Male Brown Norway rats (≥6 months old) were anesthetized (3.5% isoflurane) and placed under a surgical scope. After pupil dilation (0.5% Tropicamide), 8 µm-diameter magnetic microbeads (Bangs Laboratories, Fishers, Indiana) were injected (8 µL) into the anterior chamber using a glass-pulled micropipette connected to a manual microsyringe pump and distributed along the iridocorneal angle using a neodymium magnet to occlude the trabecular meshwork. Both eyes were injected to eliminate the confounding factor of contralateral eye effects on glial activation. Separate rats were not injected and served as normotensive controls. IOP was measured for each eye using the TonoLab rebound tonometer (iCare) on non-anaesthetized rats at baseline (average of three days’ means), 48 h after OHT, at 7 days, and weekly thereafter. Each IOP measurement is the average of 10 repetitions and was performed by operators blinded to treatment.

#### 4.3.2. Treatment

Rats were randomized into 3 groups (*n* = 5 per group) based on IOP at baseline. Twenty-four hours after microbead injection, rats were treated with 5 µL intravitreal injection of balanced salt solution (BSS, Alcon Fort Worth, TX, USA; vehicle group), or human apo-transferrin (240 µg/eye, freshly prepared in BSS; transferrin (TF) group, Sigma, St. Louis, MO, USA). Treatment was then administered weekly until rats were sacrificed (4 injections total). For injection, rats were anesthetized (3.5% isoflurane and 0.5% proparacaine hydrochloride, for general and local anesthesia, respectively). Injections were performed using a 33G needle attached to a 5 µL Hamilton syringe. Post-injection, rats received ophthalmic ointment (Systane, Alcon) on each eye. Normotensive rats were left untreated (naïve group). The study was performed by operators blinded to treatment.

#### 4.3.3. RGC Quantification

RGC counting and immunolabeling of flat-mounted retinas were performed as previously described [62]. Briefly, eyes (*n* = 10 per group) were fixed in 4% PFA for 30 min. Retinas were removed from eye cups, cryoprotected in 30% sucrose overnight, then subjected to three rounds of freeze–thaw prior to vitreous removal and incubated in blocking buffer (5% donkey serum (Jackson ImmunoResearch), 1% Triton X-100 (Merck) in 1X PBS). Retinas were incubated overnight at 4 °C with anti-RBPMS (1:200, RNA-binding protein with multiple splicing, GTX118619, Genetex, Irvine, CA, USA) diluted in blocking buffer. Retinas were rinsed in PBS and incubated for 2 h at RT with Alexa-Fluor 594 secondary antibody (1/250, Jackson ImmunoResearch) diluted in blocking buffer. Retinas were mounted ganglion-cell-layer-up on slides, then cover-slipped with Fluoromount-G (SouthernBiotech, Birmingham, AL, USA). Unbiased stereological analysis of RBPMS-positive RGCs in whole-mounted retinas using a 40× objective was performed using the optical fractionator module within StereoInvestigator (MicroBrightfield Bioscience, Williston, VT, USA). A 50 × 50 µm counting frame was used across approximately 40 sites (10%), with RBPMS-positive RGCs quantified at each sampling site. The coefficient of error (Schmitz–Hof) was maintained at 0.05 or below, ensuring sufficient sampling rate. RGC counts were obtained by masked operators.

#### 4.3.4. Axon Quantification 

Optic nerves were dissected at the level of the nerve head and fixed with 4% PFA in 0.1 M sodium cacodylate buffer, pH 7.4, and then dehydrated in graded ethanol immersions and by increasing concentrations of Araldite 502-Poly/Bed^®^ 812 (Polysciences) in propylene oxide. Following embedding in epoxy resin Poly/Bed^®^ 812, optic nerves were sectioned in 0.5 µm thick cross-sections and stained with paraphenylenediamine (PPD, Sigma). Individual axons were counted by unbiased stereology using a 100× oil-objective and the optical fractionator approach within StereoInvestigator (MBF Bioscience, Williston, VT, USA). Approximately 50 sampling sites (10%) of each optic nerve cross-section were counted. The coefficient of error (Schmitz–Hof) was <0.05, ensuring a sufficient sampling rate. Axon count estimates and axon density, based on the axon counts normalized to the cross-sectional area of the optic nerve, are reported. Axon counts were obtained by masked operators.

### 4.4. Iron Staining in Human Eyes

#### 4.4.1. Human Samples

The use of human samples adhered to the tenets of the Declaration of Helsinki and was approved by the local Ethics Committee of the Swiss Department of Health on research involving human subjects (CER-VD N°340/15). The study adhered to the ARVO statement on human subjects. Postmortem whole globes were received from the Lausanne Eye Bank with informed consent if the cornea had been stated as unsuitable for transplantation. Glaucoma eyes (*n* = 2) were obtained from a 80-year-old patient with severe glaucoma. An age-matched control eye (*n* = 1) was selected from a person without a known history of eye disease and was confirmed to be normal by examination of the retina, nerve head, and optic nerve area. Because information on the specimens was obtained through pathology records (and not through the patient records), information on treatment and IOP control was incomplete.

Eyes were fixed in 4% paraformaldehyde less than 10 h after death. Blocks of tissue ranging from 1 to 1.5 cm long and 0.5 cm wide containing the optic nerve were processed as serial 5 µm-thick paraffin sections.

#### 4.4.2. Iron Detection Staining

Paraffin human post mortem eye tissues were stained for iron by the Perls method. Briefly, Perls reaction staining was performed after deparaffinization by incubating slides in a solution containing 4% potassium ferrocyanide (Sigma) in 4% aqueous hydrochloric acid (Sigma) for 1 h at room temperature to yield a Prussian blue reaction product. Sensitivity for iron detection was enhanced by incubation in 3,3′-Diaminobenzidine Tetrahydrochloride (DAB, Vector Labs, Eurobio, Courtabouef, France) for 1 h at room temperature, producing a brown reaction product. Nucleus counterstaining was performed by incubation in a solution of 1% nuclear red (Merck) for 30 s. Stained sections were mounted in aqueous medium. Control of amplification was realized by incubation of sections with DAB without the preceding Perls reaction step. Sections were analyzed by brightfield microscopy (Zeiss, Rueil Malmaison, France) and pictures were acquired using identical exposure conditions in the optic nerve or at a similar distance from the optic nerve.

### 4.5. Statistics

Statistical analyses were performed using GraphPad Prism 8 software. Data was expressed as mean ± standard error of the mean (SEM). For multiple comparisons, non-parametric analysis was performed using Kruskal–Wallis and Dunn’s post hoc tests (ex vivo data) or using one-way ANOVA with Tukey’s post hoc test (in vivo data). For IOP statistics, repeated measures ANOVA and Tukey’s multiple comparison post hoc tests were performed. A *p* value < 0.05 was considered statistically significant.

## 5. Conclusions

In conclusion, TF appears as a promising approach for the treatment of glaucoma neuropathy by protecting retinal ganglion cells against multiple pathogenic mechanisms.

## Figures and Tables

**Figure 1 ijms-23-12753-f001:**
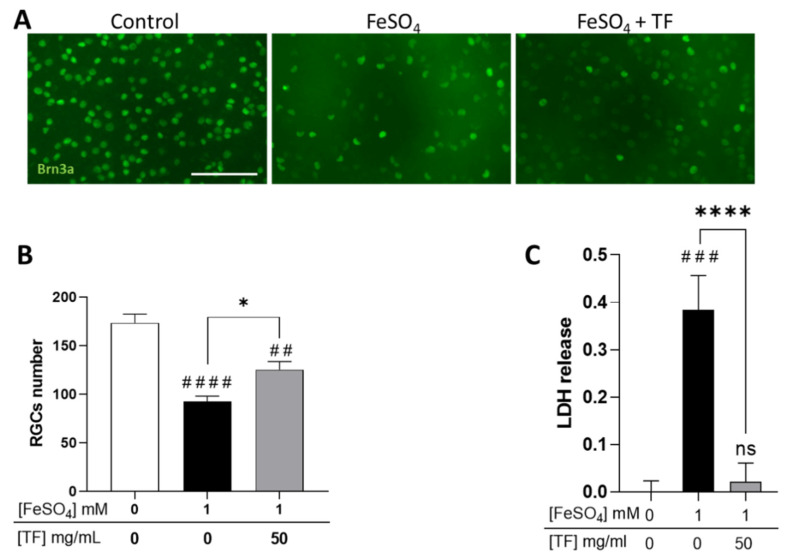
Transferrin protects RGCs in iron-treated retinal explants. (**A**) Representative images of retinal whole-mounts prepared for Brn3a immunostaining following 24 h incubation with 1 mM FeSO_4_ or transferrin (TF) (50 mg/mL) combined with FeSO_4_ and further cultured for 72 h (96 h total) in presence or not of TF. Untreated explant served as control. (**B**) Brn3a-positive cells were counted, showing that TF significantly protects against the loss of RGCs from iron overload. (**C**) LDH released by explants during 3 h to 24 h of culture was significantly increased in presence of 1 mM FeSO_4_ but absent in additive presence of TF. Data represent means ± SEM, *n* = 8–9 explants per condition. Statistical analysis was performed with Kruskal–Wallis and Dunn’s tests for multiple comparisons. ns: not significant; ## *p* < 0.01; ### *p* < 0.001; #### *p* < 0.0001 compared to control; * *p* < 0.05; **** *p* < 0.0001; compared to stress condition. Scale bar: 100 µm.

**Figure 2 ijms-23-12753-f002:**
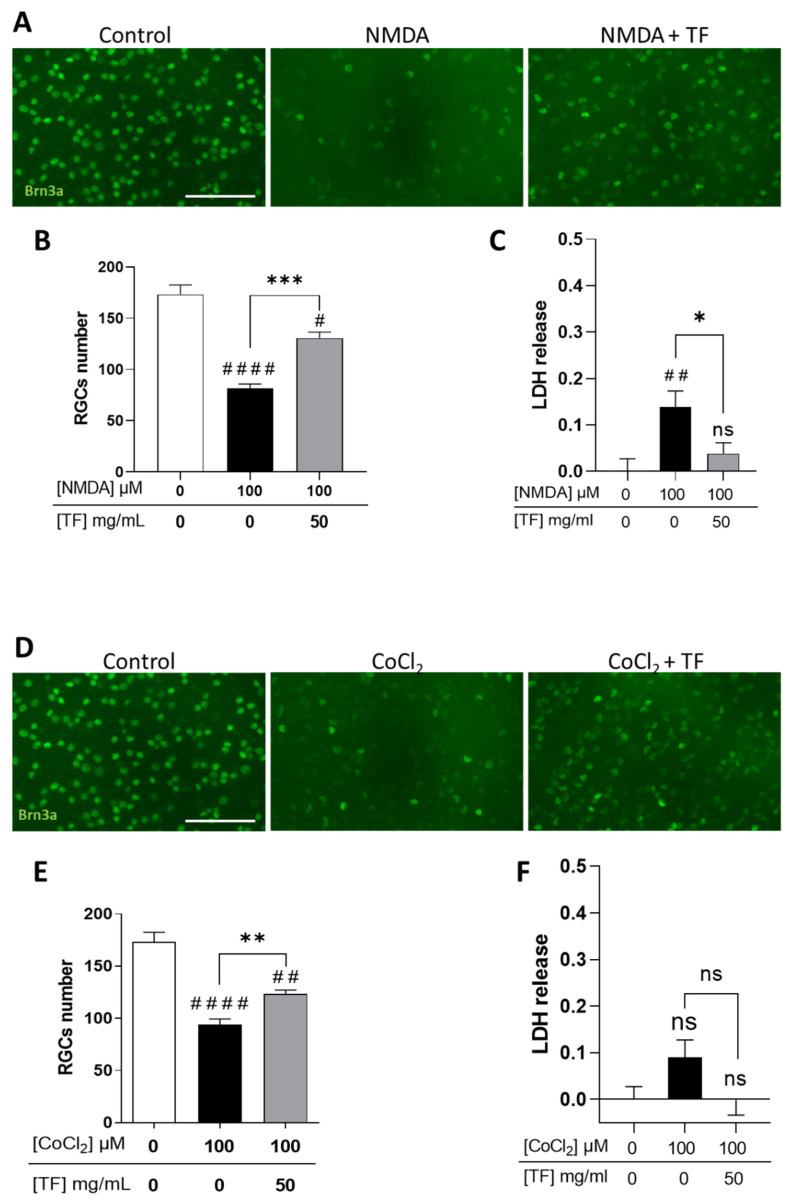
Transferrin protects RGCs in NMDA- or CoCl_2_-treated retinal explants. (**A**) Representative images of retinal whole-mounts prepared for Brn3a immunostaining following 24 h incubation with 100 µM NMDA or TF (50 mg/mL) combined with NMDA and further cultured for 72 h (96 h total) in presence or not of TF. Untreated explant served as control. (**B**) Brn3a-positive cells were counted showing that TF significantly protects RGCs against NMDA-induced cell death. (**C**) LDH released by explants during 3 h to 24 h of culture was significantly increased in presence of 100 µM NMDA and absent in additive presence of TF. (**D**) Representative images of retinal whole-mounts prepared for Brn3a immunostaining following 24 h incubation with 100 µM CoCl_2_ or TF (50 mg/mL) combined with CoCl_2_ and further cultured for 72 h (96 h total) in presence or not of TF. Untreated explant served as control. (**E**) Brn3a-positive cells were counted showing that TF significantly protects RGCs against CoCl_2_-induced cell death. (**F**) LDH released by explants during 3 h to 24 h of culturing was significantly increased in presence of 100 µM CoCl_2_ and absent in additive presence of TF. Data represent means ± SEM, *n* = 6–10 explants per condition. Statistical analysis was performed with Kruskal–Wallis and Dunn’s tests for multiple comparisons. ns: not significant; # *p* < 0.05; ## *p* < 0.01; #### *p* < 0.0001 compared to control. ns, not significant; * *p* < 0.05 ** *p* < 0.01; *** *p* < 0.001; compared to stress condition. Scale bar: 100 µm.

**Figure 3 ijms-23-12753-f003:**
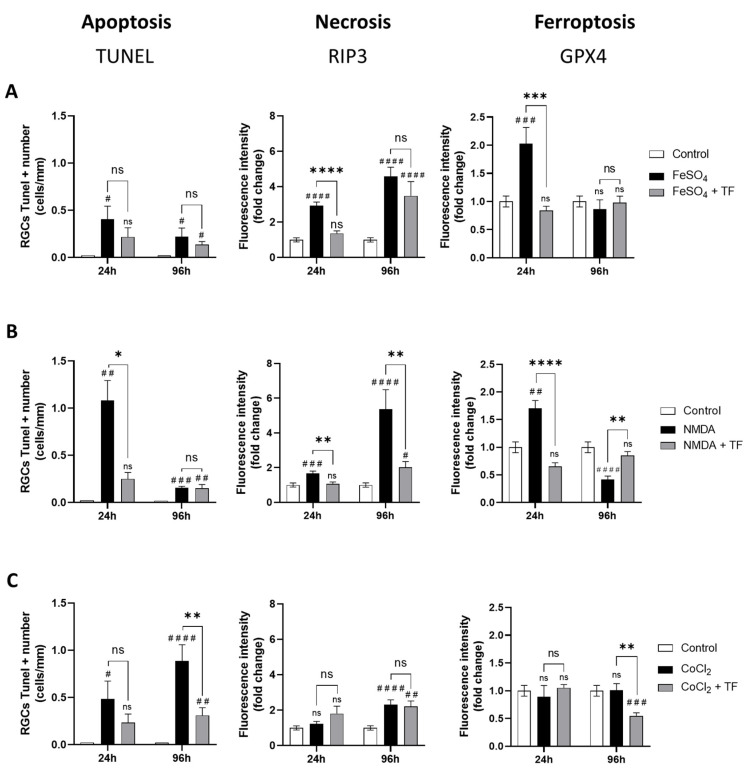
Transferrin protects RGCs against different cell-death mechanisms. Activation of apoptosis, necrosis, or ferroptosis in rat retinal explants exposed to FeSO_4_ (**A**), NMDA (**B**) or CoCl_2_ (**C**) in presence or in absence of TF was monitored by, respectively, the quantification of TUNEL-positive RGCs and by the change in RIP3 or GPX4 immunostaining intensity in RGC layer compared to controls (fold change). (**A**) FeSO_4_ induces a significant increase in RIP3 and GPX4 expression which was limited or prevented by co-treatment with TF. (**B**) NMDA increases the number of TUNEL-positive cells as well as RIP3 and GPX4 expression. TF prevents increase in all three markers. (**C**) CoCl_2_ induces a significant increase in the number of TUNEL-positive cells which was prevented by TF. Data represent means ± SEM, *n* = 4–10 explants per condition. Statistical analysis was performed with Kruskal–Wallis and Dunn’s tests for multiple comparisons. ns, not significant; # *p* < 0.05; ## *p* < 0.01; ### *p* < 0.001; #### *p* < 0.0001 compared to control. ns, not significant; * *p* < 0.05; ** *p* < 0.01; *** *p* < 0.001; **** *p* < 0.0001 compared to stress condition.

**Figure 4 ijms-23-12753-f004:**
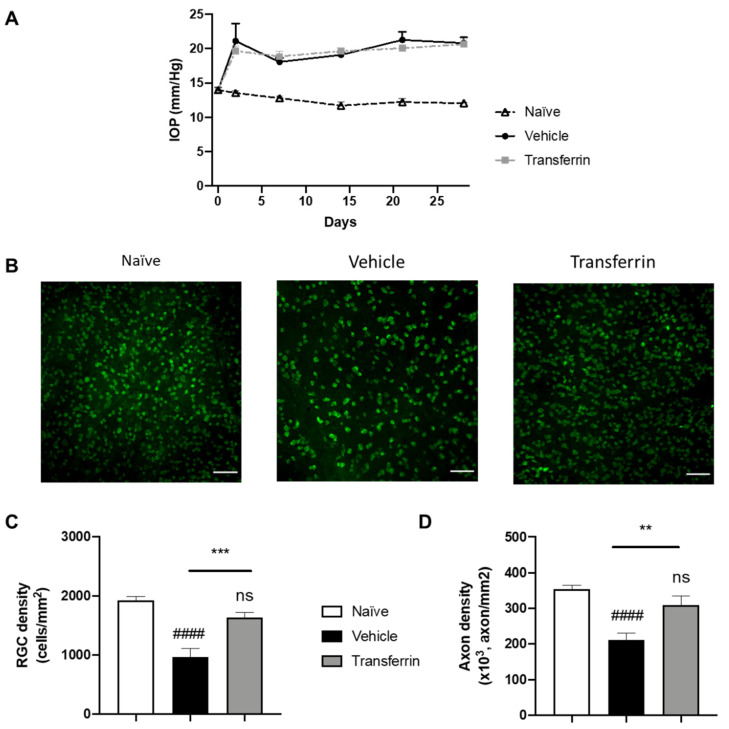
Transferrin prevents retinal ganglion cell death and axon loss in rats exposed to ocular hypertension. (**A**) Rats were injected with microbeads in the anterior chamber on day 0. IOP remained significantly elevated over the four-week period of the study beginning at 48 h after OHT when compared to normotensive naïve animals. IOPs were not statistically different in vehicle- and TF-treated groups. (**B**) Representative micrographs of flat-mounted retina sections labeled for the RGC marker RBPMS. (**C**) Mean RGC density per mm^2^ in the vehicle group was significantly reduced when compared with control animals (naïve group). TF significantly preserved RGC. (**D**) Axon density in optic nerves from vehicle-treated group was significantly lower than naïve. There was no difference in axon density between the TF treatment and naïve optic nerves. Data represent means ± SEM, *n* = 10 per group. Statistics are one-way ANOVA followed by Tukey’s multiple comparisons test. Symbols above the bars are comparisons with naïve control group. ns, not significant; #### *p* < 0.0001 compared to naïve control group. ** *p* < 0.01; *** *p* < 0.001 compared to vehicle-treated group.

**Figure 5 ijms-23-12753-f005:**
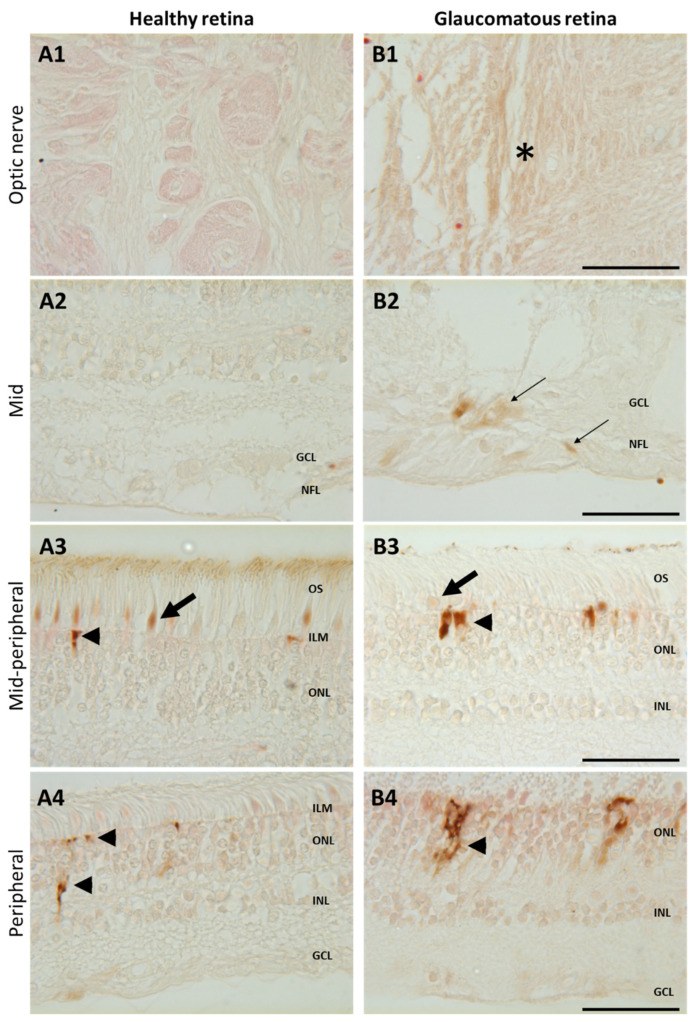
Increased iron in glaucomatous versus healthy human retinas. Representative images of enhanced Perls’ reaction realized on human healthy eyes (**A**) and glaucoma-affected eyes (**B**) observed in the optic nerve (**A1**, **B1**), in the mid- and mid-peripheral retina (**A2–3**,**B2–3**), and in the peripheral retina (**A4**,**B4**). In healthy retina, iron was localized in cone outer segments ((**A3**), arrows) and in the Müller glial cell end-feed at the inner limiting membrane (ILM) ((**A3**), arrowhead) and in processes ((**A4**), arrowhead). Glaucomatous retinas present iron deposits in optic nerve ((**B1**), asterisk) and in the nerve fiber and ganglion cell layers ((**B2**), thin arrow) and increased iron staining in Müller cells ((**B3**,**B4**), arrowheads) but decreased in cone outer segments ((**B3**), arrow). Scale bar: 50 µm. GCL: Ganglion cell layer; ILM: Inner limiting membrane; INL: Inner nuclear layer; NFL: Nerve fiber layer; ONL: Outer nuclear layer. OS: Outer segment.

**Table 1 ijms-23-12753-t001:** Bibliographic review of iron metabolism and glaucoma in primates.

Authors	Population	Data	**Ref.**
R. C. Tripathi et al.	36 patients with POAG, 18 patients with secondary glaucoma, and 33 age-matched control cataract patients	Increased aqueous humor TF levels	[11]
M. I. Vinetskaia and E. N. Iomdina	Adult Russian population (48 samples)	Higher iron content in lacrimal fluids	[37]
T. Miyahara et al.	Cynomolgus monkeys glaucoma IOP model (4 eyes)	Increase mRNA and proteins staining for CP in retina	[13]
R. H. Farkas et al.	Male juvenile cynomolgus monkeys glaucoma IOP model23 glaucoma eyes and 23 control eyes	Increase mRNA and proteins staining for CP, FT, TF in inner retinal layers	[12]
K. Stasi et al.	6 glaucomatous eyes and 6 control eyes	Increase CP staining in Müller glial cells	[35]
K. N. Engin et al.	160 patients with POAG and 31 control cataract patients	Higher serum TF levels	[42]
R. Sorkhabi et al.	22 patients with POAG and 25 age- and sex-matched cataract control patients	Higher aqueous humor hepcidin levels	[36]
S.-C. Lin et al.	South Korean participants (9632 individuals included)	Higher serum FT levels	[18]
H. J. Gye et al.	South Korean participants (164,029 individuals included)	Higher serum FT levels	[43]
M. Sarnat-Kucharczyk et al.	30 patients with POAG and 25 control cataract patients	Lower serum CP levels	[44]
B. Hohberger	12 patients with POAG, 10 patients with PEXG, and 11 control cataract patients	Lower aqueous humor iron levels only in PEXG	[38]
A. Fick et al.	22 patients with POAG and 18 control patients	Higher serum iron levels	[45]
E. Iomdina et al.	12 patients with POAG and 18 control cataract patients	Higher aqueous humor iron levels in POAG	[39]
B. Bocca et al.	35 patients with POAG and 20 age-matched controls	Higher aqueous humor iron levels in POAG	[40]
M. Aranaz et al.	17 patients with PEXG, 16 patients with POAG, and 16 age-matched cataract control patients	No differences	[41]
F. Yao et al.	31 patients with APACG and 31 non-glaucomatous patients	Higher serum iron (ferric form) in APACG	[46]

The search was performed on PubMed and Google Scholar using the keyword combinations of: ‘iron’ or ‘Fe’ or ‘transferrin’ or ‘ferritin’ or ‘ceruloplasmin’ or ‘hepcidin’ or ‘ferroportin’ or ‘hephaestin’ or ‘lipocalin2’ or ‘DMT1’ or ‘HFE’ and ‘glaucoma’ or ‘Fe’ and ‘glaucoma’. The search included all publications in peer-reviewed journals from 1992 through the date of the search (18 July 2022). The search included all types of publications (original articles, reviews, meta-analyses, case reports, etc.) and all available journals and sources. All the 229 results were checked one by one by reading summary and paper if available. Only results on humans or primates were included. Legend: APACG: Acute primary angle-closure glaucoma; CP: Ceruloplasmin; FT: Ferritin; IOP: Intra-ocular pressure; PEXG: pseudo-exfoliation glaucoma; POAG: primary open-angle glaucoma; TF: Transferrin.

**Table 2 ijms-23-12753-t002:** Primary antibodies.

Primary Antibodies	Host Species	Reacts With	Dilution	Manufacturer	References
GPX4	Rabbit	Mouse, Rat, Human	1:200	Abcam	ab125066
RIP3	Rabbit	Mouse, Rat, Human	1:100	Santa Cruz Biotechnologies	sc-374939
RIP3	Mouse	Mouse, Rat, Human	1:100	Merck	PRS2283

## Data Availability

Data is contained in the manuscript and Appendix A provided herein.

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
