# Peer review of "Neuroprotective Effects of Transferrin in Experimental Glaucoma Models"

_ijms, 2022, doi:10.3390/ijms232112753_

Round 1
Reviewer 1 Report
Youale and colleagues provided an experimental study to imply the protective role of transferrin in and animal model of glaucoma. Although the main idea of study in somehow novel, and the narrative structure of the study is quite interesting, several issues need to be considered.
1.The authors should mention clearly what is newly presented in their study compared with recently published studies in this field.
https://iovs.arvojournals.org/article.aspx?articleid=2775336
https://iovs.arvojournals.org/article.aspx?articleid=2780241
2. The sample size used in this study is quite low to a reach a definite conclusion about the role of transferrin in neuroprotection in glaucoma patients.
Author Response
Youale and colleagues provided an experimental study to imply the protective role of transferrin in and animal model of glaucoma. Although the main idea of study in somehow novel, and the narrative structure of the study is quite interesting, several issues need to be considered.
1.The authors should mention clearly what is newly presented in their study compared with recently published studies in this field.
https://iovs.arvojournals.org/article.aspx?articleid=2775336
https://iovs.arvojournals.org/article.aspx?articleid=2780241
We presented the two abstracts the reviewer is pointed out at the 2021 and 2022 ARVO annual meetings, based on unpublished data regarding our ongoing investigation of the neuroprotective potential of transferrin in ex vivo and in vivo glaucoma rat models. The extensive analyses were completed and described in this original article. This work has never been published elsewhere.
- The sample size used in this study is quite low to a reach a definite conclusion about the role of transferrin in neuroprotection in glaucoma patients.
We respectfully disagree with the reviewer's comment. Although the exploration of iron accumulation in glaucoma patients was conducted in a limited number of eyes for obvious reasons (e.g., limited sources, ethical reasons), the purpose of this experiment was to further support previously published observations.
In contrast, the ex vivo and in vivo studies to confirm the neuroprotective role of transferrin in glaucoma conditions were conducted in well-powered studies. Indeed, ex vivo experiments were conducted with sample size between 4 to 10 explants. The in vivo experiment was conducted with a sample size of 10 eyes per experimental conditions as indicated in the materials and methods section. Our results have been analysed with the greatest rigor from a statistical point of view as explained in the material and method section, with the guidance of an independent statistician.
Reviewer 2 Report
In the present manuscript, the authors have reported the neuroprotective effects of transferrin (TF) in experimental glaucoma model (ex vivo retina organ culture) and tested the key pathway hypothesized in retina isolated from glaucoma patient samples.
The authors have tested the effect of TF on many known pathways of glaucoma including adopting FeSO4-induced cytotoxicity (oxidative stress), NMDA-induced excitotoxicity, CoCl2-induced hypoxia and also employed magnetic microbead injection model of rat glaucoma. They have also used human glaucoma retinal samples obtained from the post mortem eyes to selectively identify the probable localization of focal presence of iron. The results obtained analyzing various experimental data show evidence that TF interferes with the cell death/damage pathway activation due to iron-induced oxidative stresses in glaucoma model and in patient sample. Iron is mostly localized in the optic nerve head or in the nerve fiber and the ganglion cell layers indicating a probable cause of optic nerve damage.
Overall, this is an interesting study and the results show potential role of TF to be considered as an important molecule from a therapeutic perspective to treat glaucoma. I have noticed many areas in the manuscript where necessary attention is needed from the authors to address the comments and questions I have.
Major:
1. Glaucoma is a group of neurodegenerative ocular condition and it consists of several subtypes. Is the current research dedicated solely to identify the role of FN in primary open angle glaucoma associated with significant rise of IOP?
2. Why two types of animal models have been used in the present study (Wister/Norway rats)? Which study uses retina/RGC from what animals is not clear? What are the sexes of animals used?
3. The whole manuscript has inconsistently reported use of retina or RGC. Both have been used synonymously but providing a consistent and clear description will add more value to the study.
4. Though the experimental description for the immune fluorescence states use of DAPI as the nuclear stain, no images provided with the manuscript has any indication of nuclear stain data.
5. The appropriate controls [negative control (no primary antibody) and TF only] are missing in the immunofluorescence data.
6. Brn3a is a RGC marker and this information needs to be emphasized at the very beginning before describing the data of Brn3a immunofluorescence.
7. Statistical data reporting needs to be improved throughout the manuscript and p values (Significance) need to be properly stated (eg NS in Figs 4C and D: NS with respect to which group?). Supplementary table 1: significance of ‘b’ has not been described.
8. Why show data in the supplementary figure. Inclusion of supplementary data and table to the main manuscript would be beneficial for the readers to understand the implication of the study.
9. The microbead-model of glaucoma fits well with the outflow restricted IOP elevation in glaucoma. What happens to the retina in terms of iron accumulation and metabolism in other types of glaucoma where IOP does not raise significantly (eg in normotensive glaucoma)?
10. Are those primary antibodies used in the present studies validated against rat? Adding any supportive data if available in a separate table would be valuable.
Minor:
1. Delete ‘in the world’ from the opening statement of the introduction section (P1,L33)
2. Et al, ex vivo used throughout the manuscript needs to be italicized.
Author Response
Reviewer 2
In the present manuscript, the authors have reported the neuroprotective effects of transferrin (TF) in experimental glaucoma model (ex vivo retina organ culture) and tested the key pathway hypothesized in retina isolated from glaucoma patient samples.
The authors have tested the effect of TF on many known pathways of glaucoma including adopting FeSO4-induced cytotoxicity (oxidative stress), NMDA-induced excitotoxicity, CoCl2-induced hypoxia and also employed magnetic microbead injection model of rat glaucoma. They have also used human glaucoma retinal samples obtained from the post mortem eyes to selectively identify the probable localization of focal presence of iron. The results obtained analyzing various experimental data show evidence that TF interferes with the cell death/damage pathway activation due to iron-induced oxidative stresses in glaucoma model and in patient sample. Iron is mostly localized in the optic nerve head or in the nerve fiber and the ganglion cell layers indicating a probable cause of optic nerve damage.
Overall, this is an interesting study and the results show potential role of TF to be considered as an important molecule from a therapeutic perspective to treat glaucoma. I have noticed many areas in the manuscript where necessary attention is needed from the authors to address the comments and questions I have.
Major:
- Glaucoma is a group of neurodegenerative ocular condition and it consists of several subtypes. Is the current research dedicated solely to identify the role of FN in primary open angle glaucoma associated with significant rise of IOP?
Glaucoma is a multifactorial disease involving different mechanisms associated or not with elevated IOP. In this study, we first focused on the protective role of TF against different pathophysiological mechanisms involved in RGC death without any discrimination of a specific type of glaucoma. The inducible hypertensive rat model of glaucoma was chosen to demonstrate the protective effect of TF because elevation of IOP leads to representative, robust, and continuous non-synchronized RGC death that occurred in either acute or chronic glaucoma. As discussed in the article, it cannot be excluded that deregulation of iron metabolism, that is probably not limited to the POAG form (see question 9), could be an early event in the glaucomatous neuropathy and not only secondary to cell injury/death and/or chronic inflammation. This suggests that the neuroprotective role of TF could be extended to different types of glaucoma including normotensive and hypertensive forms. Modifications have been made to the manuscript to mention the broader potential of TF outside POAG (Page 13, Lines 362-373).
- Why two types of animal models have been used in the present study (Wister/Norway rats)? Which study uses retina/RGC from what animals is not clear? What are the sexes of animals used?
The text was improved to clarify gender and strain used in the different experiments (Page 13, Lines 3942-395-403). Different strains were used according to the models already described in the literature. Thus, male Wistar rats were used for retinal explant culture as established in Daruich et al., 2019. The magnetic microbead occlusion model is a variation of the model originally published by Samsel et al., 2011 and Sappington et al., 2010, and established in Brown Norway rats.
- The whole manuscript has inconsistently reported use of retina or RGC. Both have been used synonymously but providing a consistent and clear description will add more value to the study.
Changes were made all along the text to clarify this point.
- Though the experimental description for the immune fluorescence states use of DAPI as the nuclear stain, no images provided with the manuscript has any indication of nuclear stain data.
The reviewer is correct. Most of the immunofluorescent mounting were counterstained with DAPI to help the localization of the ganglion cell layer. However, we made the decision not to include these nuclear staining images in the paper to not overload the figure. Representative images are now provided as Supplementary Data (Supplementary Figure 3).
- The appropriate controls [negative control (no primary antibody) and TF only] are missing in the immunofluorescence data.
Controls suggested by the reviewer have been performed but not shown in the manuscript. Retinal explants were incubated with TF only with no impact on RGC survival as already mentioned in the manuscript (Page 3, Line 100). Examples of immunofluorescence negative control (without primary antibody) are provided below.
Negative control staining: Rat retinal explant exposed to FeSO4 were incubated with the Alexa Fluor 647–conjugated donkey anti-goat IgG secondary antibody without the primary antibody. No signal was observed in the RGCs layer (outlining in white dots) in photograph acquired with a fluorescence microscope using identical exposure parameters of samples. DAPI staining counterstained nuclei. Scale bar: 50µm
- Brn3a is a RGC marker and this information needs to be emphasized at the very beginning before describing the data of Brn3a immunofluorescence.
The information has been emphasized on Page 2, Line 93.
- Statistical data reporting needs to be improved throughout the manuscript and p values (Significance) need to be properly stated (eg NS in Figs 4C and D: NS with respect to which group?). Supplementary table 1: significance of ‘b’ has not been described.
Reporting of the statistical data in legends of figures and in the text was improved as suggested by the reviewer. The presence of ‘b’ in Supplementary table 1 was a typing error. It has now been deleted.
- Why show data in the supplementary figure. Inclusion of supplementary data and table to the main manuscript would be beneficial for the readers to understand the implication of the study.
The main objective of the present paper was to explore the protective effect of TF on RGC survival. All the data supporting this objective were included in the manuscript. The supplementary data provide the reader with information on model establishment (e.g., choice of concentration of different stressor leading RGC cell death), and to highlight the consistency of the models used.
- The microbead-model of glaucoma fits well with the outflow restricted IOP elevation in glaucoma. What happens to the retina in terms of iron accumulation and metabolism in other types of glaucoma where IOP does not raise significantly (eg in normotensive glaucoma)?
There is no published data reporting a direct link between normotensive glaucoma (NTG) and intraretinal iron accumulation. However, there are some evidence suggesting a dysregulation of iron metabolism in normotensive glaucoma. Indeed, mutations of the Optineurin (OPTN) gene have been associated with the risk of NTG occurrence (Toda et al., 2004; Weisschuh et al., 2005), and were shown to induce a disruption in Rab8 interaction with TF receptor 1 (TFR1) leading to a defect in recycling of the TFR1/TF complex which is essential for the control of iron levels (Chi et al., 2010; Park et al., 2010). In addition, normotensive glaucoma is a multifactorial disease also characterised by vascular dysregulation and increased Endothelin-1 (ET-1) content (Cellini et al., 1997; Galassi et al., 2011). Interestingly, increased ET-1 level leads to iron accumulation (Kasztan et al., 2022). Modifications have been made to the manuscript to mention the iron dysregulation in NTG (Page 12-13, Lines 362-374).
- Are those primary antibodies used in the present studies validated against rat? Adding any supportive data if available in a separate table would be valuable.
All primary antibodies were validated by manufacturers to recognize the corresponding rat protein. As suggested by the reviewer, a table summarizing the primary antibodies (with specific references) used in the study for the immunochemistry analyses was added in Materials and Methods section (Page 14, Table 2). In addition, specific references of antibodies missing in the text were specified in the Materials and Methods section (Page 13, Line 421; Page 14, Table 2; Page 15, Line 505).
Minor:
- Delete ‘in the world’ from the opening statement of the introduction section (P1, L33)
“In the word” was deleted (Page 1, Line 33)
- Et al, ex vivo used throughout the manuscript needs to be italicized.
Changes were performed throughout the manuscript as suggested by the reviewer.
References:
Cellini, M., Possati, G.L., Profazio, V., Sbrocca, M., Caramazza, N., Caramazza, R., 1997. Color Doppler imaging and plasma levels of endothelin-1 in low-tension glaucoma. Acta Ophthalmol Scand Suppl 11–13. https://doi.org/10.1111/j.1600-0420.1997.tb00448.x
Chi, Z.-L., Akahori, M., Obazawa, M., Minami, M., Noda, T., Nakaya, N., Tomarev, S., Kawase, K., Yamamoto, T., Noda, S., Sasaoka, M., Shimazaki, A., Takada, Y., Iwata, T., 2010. Overexpression of optineurin E50K disrupts Rab8 interaction and leads to a progressive retinal degeneration in mice. Hum Mol Genet 19, 2606–2615. https://doi.org/10.1093/hmg/ddq146
Daruich, A., Le Rouzic, Q., Jonet, L., Naud, M.-C., Kowalczuk, L., Pournaras, J.-A., Boatright, J.H., Thomas, A., Turck, N., Moulin, A., Behar-Cohen, F., Picard, E., 2019. Iron is neurotoxic in retinal detachment and transferrin confers neuroprotection. Sci Adv 5, eaau9940. https://doi.org/10.1126/sciadv.aau9940
Galassi, F., Giambene, B., Varriale, R., 2011. Systemic vascular dysregulation and retrobulbar hemodynamics in normal-tension glaucoma. Invest Ophthalmol Vis Sci 52, 4467–4471. https://doi.org/10.1167/iovs.10-6710
Kasztan, M., Hyndman, K.A., Binning, E., Pollock, J.S., Pollock, D.M., 2022. Endothelin-1 via ETA receptor activation promotes renal iron deposition in murine models of iron overload. The FASEB Journal 36. https://doi.org/10.1096/fasebj.2022.36.S1.R3470
Park, B., Ying, H., Shen, X., Park, J.-S., Qiu, Y., Shyam, R., Yue, B.Y.J.T., 2010. Impairment of protein trafficking upon overexpression and mutation of optineurin. PLoS One 5, e11547. https://doi.org/10.1371/journal.pone.0011547
Samsel, P.A., Kisiswa, L., Erichsen, J.T., Cross, S.D., Morgan, J.E., 2011. A novel method for the induction of experimental glaucoma using magnetic microspheres. Invest Ophthalmol Vis Sci 52, 1671–1675. https://doi.org/10.1167/iovs.09-3921
Sappington, R.M., Carlson, B.J., Crish, S.D., Calkins, D.J., 2010. The Microbead Occlusion Model: A Paradigm for Induced Ocular Hypertension in Rats and Mice. Invest Ophthalmol Vis Sci 51, 207–216. https://doi.org/10.1167/iovs.09-3947
Toda, Y., Tang, S., Kashiwagi, K., Mabuchi, F., Iijima, H., Tsukahara, S., Yamagata, Z., 2004. Mutations in the optineurin gene in Japanese patients with primary open-angle glaucoma and normal tension glaucoma. American Journal of Medical Genetics Part A 125A, 1–4. https://doi.org/10.1002/ajmg.a.20439
Weisschuh, N., Neumann, D., Wolf, C., Wissinger, B., Gramer, E., 2005. Prevalence of myocilin and optineurin sequence variants in German normal tension glaucoma patients. Mol Vis 11, 284–287.
Reviewer 3 Report
- Keywords should appear in alphabetical order.
- Conclussions should appear at the end of the article, and be clear and concise
Author Response
Reviewer 3
- Keywords should appear in alphabetical order.
Keywords are now presented in alphabetical order as suggested by the reviewer (Page 1, Lines 29-30).
- Conclusions should appear at the end of the article, and be clear and concise
The formatting of the manuscript have been changed to add the conclusion in a separate section (Page 13, Lines 3764-387).

Round 2
Reviewer 1 Report
I would like to thank the authors for providing the responses. The manuscript is well-written and can be accepted in the present form.
Reviewer 2 Report
The revised manuscript has been improved substantially. The authors have taken adequate care to prepare responses relevant to my comments and concerns.